# Association of the Neutrophil-to-Lymphocyte Ratio with 90-Day Functional Outcomes in Patients with Acute Ischemic Stroke

**DOI:** 10.3390/brainsci14030250

**Published:** 2024-03-04

**Authors:** Licong Chen, Lulu Zhang, Yidan Li, Quanquan Zhang, Qi Fang, Xiang Tang

**Affiliations:** 1Department of Neurology, The First Affiliated Hospital of Soochow University, Suzhou 215000, China; 20215232103@stu.suda.edu.cn (L.C.); zll@suda.edu.cn (L.Z.); 20215232104@stu.suda.edu.cn (Y.L.); zhangquanquan@suda.edu.cn (Q.Z.); 2Department of Neurology, Dushu Lake Hospital Affiliated to Soochow University, Suzhou 215000, China

**Keywords:** acute ischemic stroke, inflammation, neutrophil-to-lymphocyte ratio, 90-day functional outcome

## Abstract

The neutrophil-to-lymphocyte ratio (NLR), an inflammatory marker, plays an important role in the inflammatory mechanisms of the pathophysiology and progression of acute ischemic stroke (AIS). The aim of this study was to identify the potential factors associated with functional prognosis in AIS. A total of 303 AIS patients were enrolled in this study; baseline information of each participant, including demographic characteristics, medical history, laboratory data, and 90-day functional outcome, was collected. Multivariate logistic regression analysis revealed that NLR, systolic blood pressure (SBP) and National Institutes of Health Stroke Scale (NIHSS) score were found to be independent factors for poor functional outcomes. Receiver operating characteristic (ROC) curve analysis was performed to estimate the predictive value of the NLR for 90-day functional outcome, with the best predictive cutoff value being 3.06. In the multivariate logistic regression analysis, three models were constructed: Model 1, adjusted for age, sex, SBP, and TOAST classification (AUC = 0.694); Model 2, further adjusted for the NIHSS score at admission (AUC = 0.826); and Model 3, additionally adjusted for the NLR (AUC = 0.829). The NLR at admission was an independent predictor of 90-day prognosis in patients with AIS. The risk factors related to poor 90-day functional outcomes were higher SBP, higher NLR, and a greater NIHSS score.

## 1. Introduction

Globally, stroke is the second leading cause of death and the third leading cause of death and disability [1]. Timely prediction and intervention can effectively reduce the disability rate among patients. The pathogenesis of acute ischemic stroke (AIS) involves multiple factors, including in situ thrombotic occlusion, artery-to-artery embolism, local branch occlusion, the inflammatory response, ischemia–reperfusion injury, and other contributing mechanisms [2]. Studies have shown that immune responses play an important role in the pathophysiology and progression of AIS, particularly with regard to brain injury and tissue repair [3,4,5]. In fact, inflammation and immune mechanisms can be involved throughout all stages of the disease [6].

In recent studies, novel inflammatory markers, including the neutrophil-to-lymphocyte ratio (NLR), C-reactive protein (CRP), the CRP-to-lymphocyte ratio (CLR), the platelet-to-lymphocyte ratio (PLR), and the systemic immune-inflammation index (SII), have been acknowledged as surrogate markers of systemic inflammation, and they have been valuable indicators for the diagnosis and prognosis of diverse infectious diseases, including cancer [7,8,9], heart disease [10,11], acute respiratory distress syndrome (ARDS) [12], and COVID 2019 [13]. According to recent studies, these markers can also predict the complications of stroke. The NLR is significantly associated with the clinical prognosis of stroke patients, including both patient outcomes following endovascular therapy (EVT) [14] and the risk of hemorrhagic transformation (HT) [15]. In addition, increased systemic inflammation is associated with the risk of progressive stroke [16,17] and the severity of cerebral edema early after reperfusion therapy in stroke patients [18].

The aim of this study was to identify potential factors associated with 90-day functional outcomes and explore the relationships between these factors and functional prognosis in AIS. This study focused on common clinical inflammatory markers, including NLR, CLR, SII, CRP, FIB, PLR, HCY, and WBC and ultimately validated the utility of the NLR as a reliable prognostic indicator for AIS patients. Furthermore, our study integrated clinical factors, biomarkers, and imaging to construct a highly accurate prediction model, emphasizing that prompt and effective clinical intervention can significantly improve patient outcomes.

## 2. Materials and Methods

### 2.1. Study Design and Participants

This was a single-center retrospective cohort study of all consecutive AIS patients admitted to our stroke unit between July 2021 and October 2022. The inclusion criteria were established based on the confirmation of AIS diagnosis through diffusion-weighted imaging (DWI). The exclusion criteria were as follows: (1) had fever or an infectious disease on admission or a history of immune system disease, (2) lacked complete imaging, laboratory, or follow-up data, (3) had a life expectancy of less than 3 months or were unable to complete the study for other reasons, and (4) were unable to comprehend or adhere to study protocols or follow-up procedures due to mental, cognitive, or emotional impairments. Finally, a total of 303 AIS patients fulfilled the inclusion criteria and were included in the study (Figure 1).

This study was conducted according to the guidelines of the Declaration of Helsinki and approved by the Research Ethics Committee of the First Affiliated Hospital of Soochow University.

### 2.2. Data Collection

Basic information on each participant who met the inclusion and exclusion criteria was collected, including age, sex, systolic blood pressure (SBP), diastolic blood pressure (DBP), medical history (including hypertension, diabetes mellitus, smoking, alcoholic habit, history of stroke, atrial fibrillation, and other heart diseases), and clinical data on admission (including relevant laboratory indicators, stroke severity at admission as measured by the NIHSS score, stroke etiology established based on the TOAST classification, computed tomography perfusion imaging (CTP)-positive, HT, and thrombolytic treatment).

The laboratory data included triglyceride (TG), total cholesterol (TC), low-density lipoprotein cholesterol (LDL-C), albumin, blood platelet count, creatinine, uric acid, hemoglobin A1c, fasting blood glucose (FBG), fibrinogen (FIB), homocysteine (HCY), white blood cell (WBC), CRP, NLR, CLR, PLR and SII. The NLR was calculated as the neutrophil count/lymphocyte count. The CLR was calculated as the CRP/lymphocyte count. The PLR was calculated as the platelet count/lymphocyte count. The SII was calculated as the platelet count × neutrophil count/lymphocyte count. All blood samples were obtained within 24 h of admission.

The modified Rankin scale (mRS) scores ranged from 0 to 6, with a score of 0 indicating no symptoms, a score of 1 indicating no clinically significant disability, a score of 2 indicating slight disability, a score of 3 indicating moderate disability, a score of 4 indicating moderately severe disability, a score of 5 indicating severe disability, and a score of 6 indicating death. We defined mRS scores of 0–1 at 90 days after AIS onset as excellent functional outcomes and mRS scores of 2–6 as poor functional outcomes.

### 2.3. Statistical Tests

SPSS version 26.0 (SPSS, Inc., Chicago, IL, USA) was used for the statistical analysis. The patients were categorized into two groups based on their mRS score at 90 days: those with excellent functional outcomes (mRS: 0–1) and those with poor functional outcomes (mRS: 2–6). The normality of the distribution was assessed using the Shapiro–Wilk test. Continuous variables are presented as the mean ± standard deviation (SD), while nonnormally distributed variables are presented as medians and interquartile ranges. Categorical data were examined using the chi-squared test. The difference between two groups was analyzed using a *t*-test for normally distributed continuous variables or the Mann–Whitney *U* test for continuous variables that do not follow a normal distribution. Univariate logistic regression and multivariate logistic regression analyses were also conducted to evaluate the association between the NLR and 90-day clinical outcomes, and odds ratios (ORs) and 95% confidence intervals (95% CIs) were calculated. Receiver operating characteristic (ROC) curve analysis was performed to estimate the predictive value of the NLR for 90-day functional outcomes, and the optimal cutoff value was determined based on the maximum Youden index. In the multivariate logistic regression analysis, three models were constructed: Model 1, adjusted for age, sex, SBP, and TOAST classification; Model 2, further adjusted for the NIHSS score at admission; and Model 3, additionally adjusted for the NLR.

## 3. Results

### 3.1. Demographics and Baseline Characteristics of All Participants

As shown in Table 1, significant differences between the two groups were described by age, and patients with poor functional outcomes were older than patients with excellent functional outcomes (t = 2.10; *p* < 0.05). Patients with higher SBP (t = 2.97, *p* < 0.05) and higher TC (t = 2.15, *p* < 0.05) were more likely to suffer from poor functional outcomes. Poor functional outcome patients showed higher NIHSS scores (Z = 8.47, *p <* 0.01) and a higher NLR (Z = 5.43, *p* < 0.01). Another association was found between the two groups according to the TOAST classification (χ^2^ = 17.46, *p* < 0.01), and patients who were CTP negative (χ^2^ = 10.36, *p* < 0.01) seemed to have poor functional outcomes.

### 3.2. Comparison of Derived Blood Lymphocyte Parameters

As shown in Figure 2 and Table 2, ROC curve analysis revealed that the cutoff levels of NLR [AUC = 0.717 (0.648 to 0.786)], CLR [AUC = 0.697 (0.625 to 0.768)], SII [AUC = 0.694 (0.626 to 0.763)], CRP [AUC = 0.663 (0.587 to 0.738)], FIB [AUC = 0.661) (0.589 to 0.734)], PLR [AUC = 0.639 (0.567 to 0.710)], HCY [AUC = 0.624 (0.550 to 0.698)], and WBC [AUC = 0.622 (0.547 to 0.697)] were 3.06, 3.87, 769.83, 4.44, 3.03, 125.09, 11.15, 7.43. The NLR had a significantly greater AUC than did the CLR, SII, CRP, FIB, PLR, HCY and WBC in predicting 90-day functional outcomes in patients with AIS.

### 3.3. Analysis of Risk Factors Associated with Unfavorable Prognosis in AIS Patients

To avoid the influence of multiple comparisons, we chose *p* < 0.05 in Table 1 when performing multivariate logistic regression analysis. We selected the NLR, which has the most influential prognostic value among inflammatory factors. The three variables (NIHSS score, TOAST classification and NLR) with the most significant associations, SBP, TC and CTP-positivity were combined with age and sex and used to construct a prediction scale using a multivariate logistic model. As shown in Table 3, the SBP (OR = 1.015, *p* < 0.01), NLR (OR = 1.145, *p* < 0.01) and NIHSS score (OR = 1.241, *p* < 0.01) showed the most significant associations with poor functional outcomes according to multivariate statistical analysis, and the other variables, including age, sex, and TOAST classification, showed no associations.

### 3.4. Comparison of Functional Outcomes of AIS Patients with High or Low NLR

A ROC curve was drawn to estimate the predictive value of the NLR for 90-day poor functional outcomes. We observed that the area under the curve was 0.7 (95% CI 0.636–0.765), and the best predictive cutoff value was 3.06, with a sensitivity of 70.3% and a specificity of 67.0%. As shown in Figure 3, we divided patients into two groups around the cutoff value, and the scores on mRS are shown for the patients in the two groups who had data for the primary outcome.

### 3.5. Comparison of Various Models in Predicting 90-Day Functional Outcome with AIS

A ROC analysis was performed to examine the accuracy of the prediction scale of the 90-day mRS. As shown in Figure 4, SBP, TOAST classification, NIHSS score and NLR together showed a significantly high AUC (area under the ROC curve) of 0.829, which indicated the higher accuracy of the 90-day functional outcome prediction scale. The AUC of the predictions based on SBP and TOAST classification (green curve, AUC = 0.694) and on the SBP, TOAST classification and NIHSS score (orange curve, AUC = 0.826) are also presented. Age and sex effects were included in the multivariate logistic model to construct prediction scales.

## 4. Discussion

In this study, which included 303 patients with AIS, we found potential clinical risk factors for 90-day poor functional outcomes following AIS, including older age, higher SBP, higher TC, higher NIHSS score, TOAST classification, CTP-positive and a higher NLR. Among the various inflammatory markers examined, such as CLR, SII, CRP, FIB, SCY, WBC and PLR, the NLR exhibited a superior ability to predict the 90-day functional outcome, with an area under the curve of approximately 0.717 and an optimal cutoff value of 3.06. Furthermore, the NLR, age, sex, SBP, NIHSS score and TOAST classification substantially enhanced the accuracy of the prediction of functional outcome in patients with AIS at 90 days.

The majority of studies conducted both domestically and internationally have consistently proven that hypertension is the most important risk factor for cerebral infarction. Many studies have shown that elevated blood pressure is associated with adverse outcomes in patients with AIS who receive intravenous thrombolysis (IVT) or EVT [19,20,21]. Our study also indicates that SBP serves as an independent risk factor for poor functional outcomes in AIS patients, with a higher SBP correlating to a greater likelihood of experiencing unfavorable clinical outcomes.

A few studies have shown that elevated TC levels are associated with outcomes of ischemic stroke. A study involving 513 patients with AIS found that post-stroke mortality was negatively correlated with TC, with higher TC being a protective factor for post-stroke prognosis [22]. Our study also indicates an association between TC and stroke prognosis, although it does not act as an independent risk factor. The NIHSS score is the predominant clinical scale utilized for evaluating stroke severity. Generally, a higher NIHSS score is related to a more severe stroke. Multiple studies have shown a strong association between the NIHSS score upon admission and 90-day functional outcomes [23,24]. Our study confirms this finding. The TOAST classifies cerebral infarction into five distinct types based on the etiology of the condition [25]. The stratification of risk factors, treatment selection and prognostic assessment for various types of cerebral infarction can be facilitated according to the TOAST classification.

CTP is a medical imaging method, enabling the estimation of irreversible ischemic core damage as well as potentially salvageable ischemic penumbra [26]. CTP may aid in the selection of EVT and the prediction of the prognosis of stroke patients. The DAWN [27] and DEFUSE 3 [28] trials have demonstrated the efficacy of EVT in patients with stroke occurring more than 6 h prior, as these patients exhibit a favorable CTP penumbral pattern. However, our study revealed that CTP is associated with functional outcomes in AIS patients but is not an independent risk factor in the multivariate logistic regression analysis of AIS patients. One potential explanation is that the prognosis of AIS patients is influenced by numerous factors such as age, lesion extent, early treatments, and rehabilitation measures. Therefore, relying solely on CTP may not accurately predict prognosis. In addition, it may be related to different subjects. Our subjects received various treatments, including IVT, EVT or antiplatelet therapy alone.

The majority of strokes are caused by atherosclerosis, which leads to stenosis or occlusion of the cerebellum. This subsequently results in ischemia and hypoxia in brain tissue, leading to damage in corresponding functional areas. In recent years, growing evidence has suggested that inflammation plays a pivotal role in the initiation and progression of stroke [29,30]. After ischemia, the damaged brain tissue triggers a local inflammatory response, leading to the release of inflammatory mediators. Neutrophils, monocytes, and other immune cells accumulate and secrete metalloproteinases (MMPs), perforin, cytokines, and neutrophil extracellular traps (NETs), thereby causing damage to brain tissue. Simultaneously, thrombin is activated, disrupting endothelial barrier function and initiating complement system activation [4,31]. Blood analysis, the most commonly used clinical test, has significant value in determining the prognosis of stroke. By incorporating the ratio of various inflammatory indicators, such as the NLR, PLR, and CLR, this approach has been shown to provide greater predictive value than relying solely on individual inflammatory markers [32].

Previous studies have demonstrated the association between the NLR and stroke prognosis. The initial investigation of the relationship between the NLR and short-term mortality in stroke patients demonstrated that the NLR serves as an independent prognostic indicator of short-term mortality in AIS patients, with higher NLRs significantly associated with increased stroke-related mortality [33,34,35,36]. Min-Su Kim et al. discovered a similar finding, indicating an optimal NLR threshold value of 2.09 [37]. Our study also provided evidence that the NLR is an independent factor for 3-month clinical functional outcomes in patients with AIS. Furthermore, an increased NLR may also predict infarction size irrespective of its etiology [38]. Lattanzi et al. reported that patients with AIS and a higher NLR at admission exhibited a greater propensity for early neurological deterioration (END), with an optimal NLR threshold value of 6.4 [39]. Furthermore, multiple studies have substantiated the association between an increased NLR and HT in patients with AIS. Goyal et al. reported that the NLR was a significant independent predictor of symptomatic intracranial hemorrhage (sICH) and 3-month mortality in patients with large vessel occlusion who underwent EVT [40]. Similarly, Milena Switonska et al. also discovered that NLR at admission can accurately predict sHT in AIS patients undergoing revascularization [41]. Slaven Pikija et al. confirmed this conclusion and highlighted that the critical value of ICH is 3.89 [42]. Additionally, a high NLR is associated with an increased likelihood of stroke complications, including poststroke depression [43], stroke-associated pneumonia [44,45], delirium after stroke [46] and poststroke cognitive impairment [47,48].

The relationship between inflammation, including systemic and intravascular inflammation, and acute ischemic stroke is currently a focal point of attention. Our study focused on common clinical inflammatory markers such as NLR, CLR, SII, CRP, FIB, PLR, HCY and WBC, ultimately finding that the NLR exhibited the strongest correlation with a poor prognosis following stroke. Our study also encompassed clinical factors, laboratory indicators, including inflammation markers, and imaging in order to identify risk factors that impact the prognosis of AIS and establish a predictive model. The present model enhances the accuracy of predicting functional outcomes during the early stages of AIS, facilitating timely intervention and reducing disability rates later in disease progression. Combined with the findings of previous studies and our study, these findings further substantiated the role of immune inflammation in stroke.

This study has several limitations. First, our study was a single-center retrospective study, which resulted in the exclusion of a large number of patients and potential selection bias. Furthermore, our study did not differentiate between various reperfusion treatments, such as IVT or EVT, nor did it consider the potential impact of END on patients. Moreover, we did not include information regarding poststroke complications, which could have contributed to adverse clinical outcomes. Additionally, dynamic changes in the NLR were not monitored. Future research should further address the shortcomings mentioned above.

## 5. Conclusions

Our study demonstrated that the NLR was an independent predictor of 90-day prognosis in patients with AIS and identified potential factors associated with 90-day functional outcomes; these factors were utilized to construct highly accurate prediction models. The NLR may serve as a simple and low-cost marker for predicting functional outcomes in patients with AIS.

## Figures and Tables

**Figure 1 brainsci-14-00250-f001:**
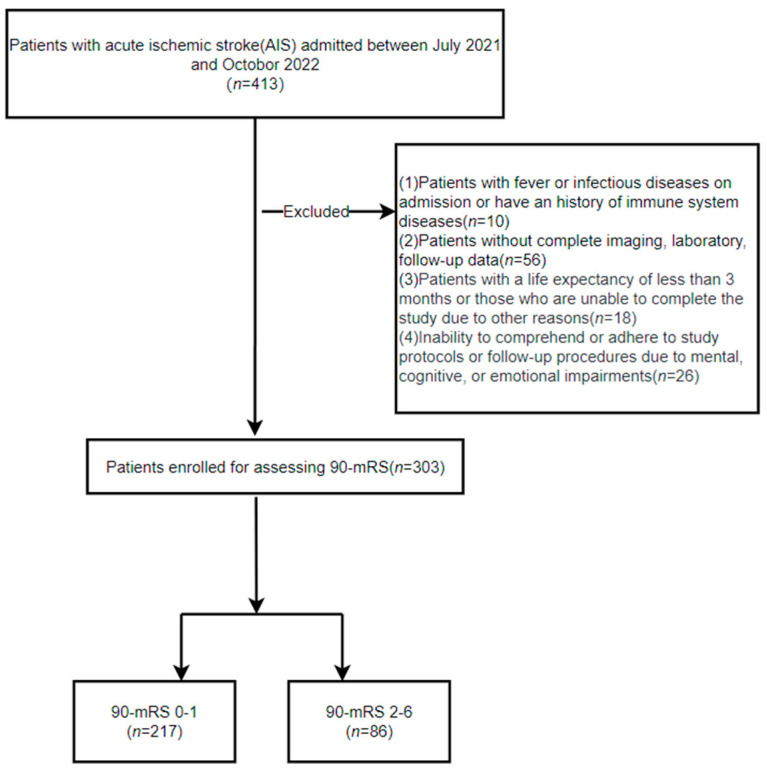
Details on study recruitment. AIS: acute ischemic stroke; mRS: modified Rankin scale.

**Figure 2 brainsci-14-00250-f002:**
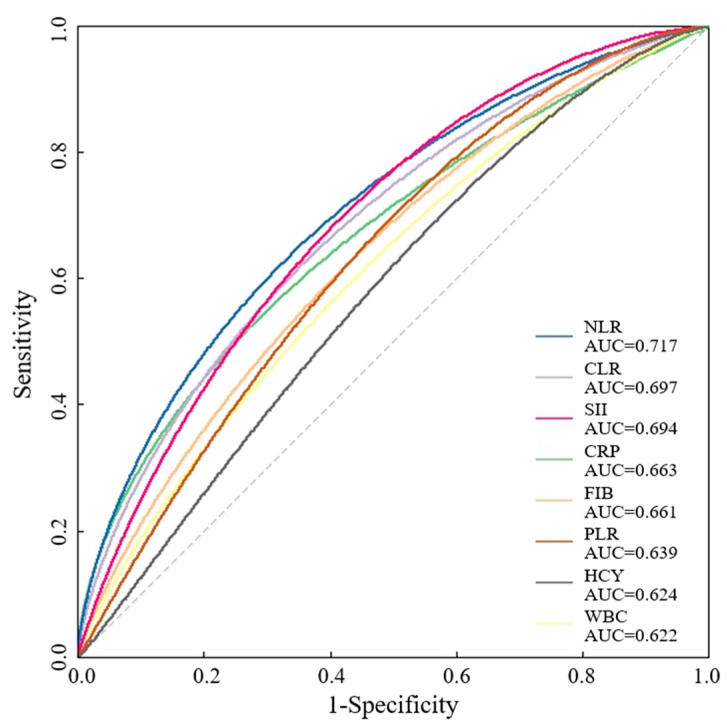
ROC curves of the derived blood inflammatory parameters to predict 90-day functional outcomes.

**Figure 3 brainsci-14-00250-f003:**
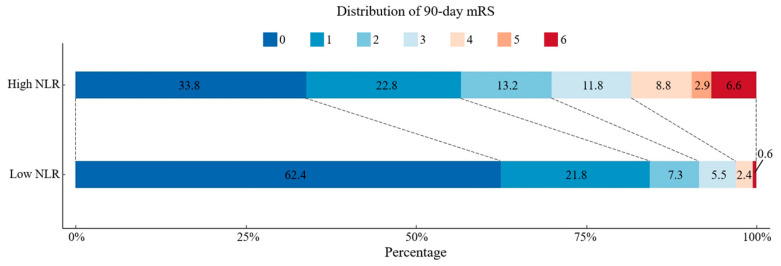
Distribution of functional outcomes at 90 days in patients with high NLR and low NLR. A high NLR is defined as an NRL greater than or equal to 3.06, and a low NLR is defined as an NLR less than 3.06. NLR: the neutrophil-to-lymphocyte ratio; mRS: modified Rankin scale.

**Figure 4 brainsci-14-00250-f004:**
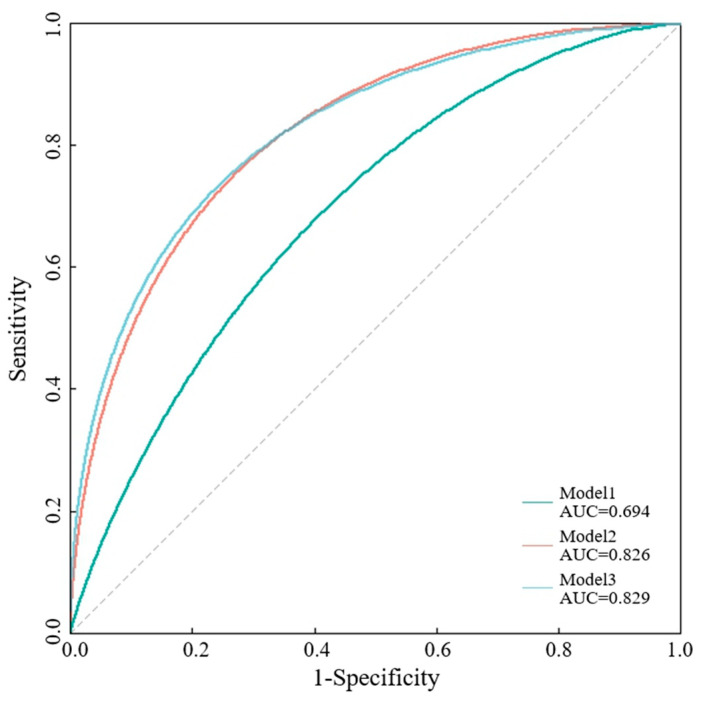
ROC curves generated for stroke with 90-day poor functional outcomes. ROC curve is generated for SBP, TOAST classification, NIHSS score and NLR (age and sex effects included) with 90-day poor functional outcomes (blue line, AUC = 0.829) based on multiple logistic regression. ROC curves are generated for SBP and TOAST classification (green, AUC = 0.694), SBP, TOAST classification and NIHSS score (orange, AUC = 0.826) with age and sex effects included.

**Table 1 brainsci-14-00250-t001:** Clinical baseline characteristic of AIS patients according to 90-day mRS.

Demographic and Clinical Data	Excellent Functional Outcome (*n* = 217)	Poor Functional Outcome (*n* = 86)	t/Z/χ^2^	*p*
Age (years)	63.91 ± 13.02	67.30 ± 11.89	t = 2.10	0.037 *
Sex male/female	149/68	56/30	χ^2^ = 0.35	0.552
SBP † (mmHg)	146.66 ± 19.88	154.51 ± 22.68	t = 2.97	0.003 *
DBP † (mmHg)	83.52 ± 11.86	84.64 ± 12.83	t = 0.72	0.470
History of hypertension yes/no	146/71	56/30	χ^2^ = 0.13	0.719
History of diabetes yes/no	63/154	23/63	χ^2^ = 0.16	0.690
Alcoholic habit yes/no	42/175	14/72	χ^2^ = 0.39	0.534
Smoking yes/no	55/162	19/67	χ^2^ = 0.35	0.552
History of AF † yes/no	20/197	8/78	χ^2^ = 0.00	0.981
Other heart diseases yes/no	12/205	4/82	χ^2^ = 0.10	0.758
Previous stroke yes/no	38/179	14/72	χ^2^ = 0.07	0.798
TG † (mmol/L)	[1.37 (1.03, 1.76)]	[1.36 (0.99, 2.04)]	Z = 0.13	0.900
TC † (mmol/L)	4.40 ± 1.05	4.69 ± 1.14	t = 2.15	0.032 *
LDL-C † (mmol/L)	2.73 ± 0.98	2.89 ± 1.09	t = 1.27	0.205
Albumin (g/L) Prealbumin (g/L)	39.11 ± 3.44 238.31 ± 50.48	39.68 ± 3.72 234.57 ± 63.79	t = 1.28 t = 0.49	0.202 0.628
Blood platelet (10^9^/L)	[208 (172, 249)]	[199 (165, 256)]	Z = 0.65	0.514
Creatinine (µmol/L)	[69.1 (58.8, 77.6)]	[71.6 (57.8, 83.9)]	Z = 1.45	0.149
Uric acid (µmol/L)	329.39 ± 96.65	347.08 ± 122.05	t = 1.20	0.231
FBG † (µmol/L)	[5.38 (4.77, 6.74)]	[5.75 (4.98, 7.44)]	Z = 1.93	0.054
CRP † (mg/L)	[2.46 (0.95, 8.05)]	[7.49 (2.58, 15.36)]	Z = 4.518	0.000 *
CLR † HCY † (µmol/L) WBC † (10^9^/L)	[1.38 (0.58, 4.90)] [9.90 (8.20, 12.10)] [6.97 (5.61, 8.54)]	[5.11 (1.61, 10.64)] [11.75 (9.60, 13.75)] [8.18 (6.60, 10.00)]	Z = 5.216 Z = 3.314 Z = 3.371	0.000 * 0.001 * 0.001 *
NLR †	[2.62 (1.94, 3.99)]	[3.93 (2.83, 6.01)]	Z = 5.43	0.000 *
SII † PLR † FIB † (g/L)	[529.74 (373.78, 841.03)] [124.64 (92.75, 171.93)] [2.80 (2.30, 3.32)]	[847.39 (531.30, 1375.46)] [149.77 (113.44, 202.28)] [3.22 (2.69, 3.95)]	Z = 4.555 Z = 3.140 Z = 3.940	0.000 * 0.002 * 0.000 *
Hemoglobin A1c ‡ (%)	[6.1 (5.6, 7.3)]	[6.1(5.7,7.2)]	Z = 0.38	0.703
NIHSS † score	[2 (1, 5)]	[7 (4, 11)]	Z = 8.47	0.000 *
TOAST classification †	98/23/80/16/0	59/10/12/5/0	χ^2^ = 17.46	0.001 *
CTP † -positive yes/no ^#^	102/64	55/11	χ^2^ = 10.36	0.006 *
Thrombolytic yes/no	53/164	28/58	χ^2^ = 2.08	0.149
HT † yes/no †	5/212	2/84	χ^2^ = 0.88	0.831

Continuous data are shown as the mean ± SD, minimum and maximum values in patients with statistical significance based on two sample *t* tests. Categorical data differences are represented with statistical significance based on the chi-square test (χ^2^ and *p*) or Fisher’s exact test (Z and *p*). * *p* < 0.05. † SBP: systolic blood pressure; DBP: diastolic blood pressure; AF: atrial fibrillation; TG: triglyceride; TC: total cholesterol; LDL-C: low-density lipoprotein cholesterol; FBG: fasting blood glucose; CRP: C-reactive protein; CLR: the CRP-to-lymphocyte ratio; HCY: homocysteine; WBC: white blood cell; NLR: the neutrophil-to-lymphocyte ratio; SII: the systemic immune-inflammation index; PLR: the platelet-to-lymphocyte ratio; FIB: fibrinogen; TOAST refers to five classifications: (1) large-artery atherosclerosis, (2) cardioembolism, (3) small-vessel occlusion, (4) stroke of other determined etiology, and (5) stroke of undetermined etiology. NIHSS: National Institutes of Health Stroke Scale; CTP: computed tomography perfusion imaging; HT: hemorrhagic transformation. ‡ We defined modified Rankin scale (mRS) scores of 0–1 at 90 days after AIS onset as excellent functional outcomes and mRS scores of 2–6 as poor functional outcomes. A total of 217 patients with excellent functional outcomes and 86 patients with poor functional outcomes underwent HCY tests, while 208 patients with excellent functional outcomes and 83 patients with poor functional outcomes underwent hemoglobin A1c tests. ^#^ A total of 166 patients with excellent functional outcomes and 66 patients with poor functional outcomes underwent CTP tests.

**Table 2 brainsci-14-00250-t002:** ROC analysis results of the derived blood lymphocyte parameters.

	AUC-ROC	95% CI	Cut-Off Level	*p* Value
NLR	0.717	0.648, 0786	3.06	<0.001
CLR	0.697	0.625, 0.768	3.87	<0.001
SII	0.694	0.626, 0.763	769.83	<0.001
CRP	0.663	0.587, 0.738	4.4	<0.001
FIB	0.661	0.589, 0.734	3.03	<0.001
PLR	0.639	0.567, 0.710	125.09	<0.001
HCY	0.624	0.550, 0.698	11.15	0.002
WBC	0.622	0.547, 0.697	7.43	0.002

NLR: the neutrophil-to-lymphocyte ratio; CLR: the CRP-to-lymphocyte ratio; SII: the systemic immune-inflammation index; CRP: C-reactive protein; FIB: fibrinogen; PLR: the platelet-to-lymphocyte ratio; HCY: homocysteine; WBC: white blood cell.

**Table 3 brainsci-14-00250-t003:** Multivariable logistic regression model for predicting patients with 90-day mRS.

Variables	Odd Ratio	95% CI	*p* Value
Age	1.016	0.99, 1.05	0.313
Sex	0.903	0.46, 1.77	0.766
SBP (mmHg)	1.015	1.00, 1.03	0.043
NLR	1.148	1.04, 1.27	0.008
NIHSS score	1.255	1.16, 1.36	0.000
Large-artery atherosclerosis	1.458	0.36, 5.96	0.599
Cardio-embolism	0.524	0.10, 2.90	0.459
Small-vessel occlusion	0.467	0.10, 2.10	0.320
TC	1.260	0.95, 1.67	0.104
CTP-positive	0.550	0.31, 1.43	0.221
CTP-negative	0.775	0.35, 1.74	0.536

SBP: systolic blood pressure; NLR: the neutrophil-to-lymphocyte ratio; NIHSS: National Institutes of Health Stroke Scale; TC: total cholesterol; CTP: computed tomography perfusion imaging.

## Data Availability

The data presented in this study are available in article.

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
