# Peer review of "Association of the Neutrophil-to-Lymphocyte Ratio with 90-Day Functional Outcomes in Patients with Acute Ischemic Stroke"

_brainsci, 2024, doi:10.3390/brainsci14030250_

Round 1
Reviewer 1 Report
Comments and Suggestions for Authors
Dear Editor,
I have read the manuscript draft titled “Association of the Neutrophil-to-lymphocyte Ratio with 90-day Functional Outcomes in Patients with Acute Ischemic Stroke” with great interest, which has been submitted for publication in BS. The article presents an original study that focuses on the role of several inflammatory markers in predicting functional outcome after acute ischemic stroke. The study concludes that NLR can be a reliable outcome predictor.
However, I would like to raise some concerns regarding the study. Firstly, it would be beneficial to define the time point of acquisition of the inflammatory biomarkers (1st day, 2nd day…?). Secondly, it would be helpful if all abbreviations were described when initially used, and then consistently used throughout the article. Additionally, I would suggest removing lines 118 - 132, and consistently reporting the results (ie. AUC). It would be helpful to change the word “Drinking” in Table 1, and check the meaning of P=1.231 in Table 1. Furthermore, I recommend revising the table captions in Table 1.
Moreover, I suggest presenting the results in lines 164-170 in a new table. Lastly, I would like to confirm whether it is “0.05” or “0.005” in line 176, and request information regarding the ranges for “high” and “low” NLR groups.
I hope you find these suggestions helpful, and I recommend that the manuscript undergoes a major revision.
Best regards.
Comments on the Quality of English LanguageNo major issues detected.
Reviewer 2 Report
Comments and Suggestions for Authors
I have read with pleasure this insightful paper by Licong Chen and colleagues, which highlights the predictive value of the NLR at 3 months in patients with ischemic stroke.
The study is methodologically well-designed, and the writing is generally clear and devoid of significant issues.
Minor suggestions for improvement:
Methods
- In the inclusion criteria, there is a reference to DWI (brain MRI). However, throughout the rest of the article, CT perfusion is discussed. Could you please clarify this discrepancy?
- Line 82: It might be more precise to use "alcoholic habit" instead of the overly general term "drinking."
- Line 91: I'm not sure about the term "CRP count." Perhaps consider revising it for clarity.
- Line 109: The term "nonparametrically distributed" is unclear; it likely refers to "continuous variables that do not follow a normal distribution."
- Lines 118-131: This appears to be an oversight; please consider removing this section.
Results
- This section is challenging to read due to the extensive amount of text within parentheses. While presenting data in this manner is not incorrect, it might be advisable to remove measures (such as means and standard deviations), which can easily be found in the table. Instead, retaining test values (t, z, chi) and p-values in parentheses would be appropriate.
- Please revise the caption for Table 1.
- Some significant p-values in Table 1 lack asterisks (*).
Discussion
- Lines 243-262: The discussion appropriately references previous studies and their findings, but there is a lack of comparison with the current study. If a previous study found that a variable predicts the outcome, it is essential to determine whether the current study confirms or refutes this finding. If the confirmation is lacking, a discussion on the reasons for this discrepancy (differences between the previous study and the current one) is necessary.
- Line 247: The discussion on total cholesterol as a predictive factor for ischemic stroke is unclear. The article primarily addresses another topic, specifically the role total cholesterol might play in predicting the prognosis of a patient three months after having a stroke.
Reviewer 3 Report
Comments and Suggestions for Authors
Chinese scientists assessed the relationship between NLR and 90-day functional outcomes in AIS.
The manuscript requires re-editing, and I am attaching my comments below:
1. there are many similar studies in the literature, so in the introduction, the authors should clearly state what is new in their study. Additionally, the authors should explain why they focused on NLR if the literature describes many similar indicators.
2. I have no critical comments about the patient recruitment process. A small note concerns the spelling of "n" in Fig. 1. It would be good to write these letters in italics.
3. I ask the authors to check whether all abbreviations are explained.
4. How were the patients included in the study treated - IVT or MT or both techniques?
5. Please provide a reference for the division of patients according to mRS.
6. The results are presented chaotically, the results should be divided into smaller subchapters and the reader should be properly introduced to the discussed statistical analysis.
7. I emphasize that in the discussion, the authors should again demonstrate what is innovative in their study.
8. Please expand the literature with the following:
https://pubmed.ncbi.nlm.nih.gov/33114150/
https://www.mdpi.com/1660-4601/20/2/898
Round 2
Reviewer 1 Report
Comments and Suggestions for Authors
Dear Editor,
The authors addressed all limitations adequately. The current draft could be considered for publication.
Best regards
Reviewer 3 Report
Comments and Suggestions for Authors
The authors have adequately addressed the comments made by the reviewer in the revised manuscript.